# DeMo: Decoupling Motion Forecasting into Directional Intentions and Dynamic States

**Bozhou Zhang**     **Nan Song**     **Li Zhang**[*]
School of Data Science, Fudan University

https://github.com/fudan-zvg/DeMo

## Abstract

Accurate motion forecasting for traffic agents is crucial for ensuring the safety and efficiency of autonomous driving systems in dynamically changing environments. Mainstream methods adopt a one-query-one-trajectory paradigm, where each query corresponds to a unique trajectory for predicting multi-modal trajectories. While straightforward and effective, the absence of detailed representation of future trajectories may yield suboptimal outcomes, given that the agent states dynamically evolve over time. To address this problem, we introduce **DeMo**, a framework that decouples multi-modal trajectory queries into two types: mode queries capturing distinct directional intentions and state queries tracking the agent's dynamic states over time. By leveraging this format, we separately optimize the multi-modality and dynamic evolutionary properties of trajectories. Subsequently, the mode and state queries are integrated to obtain a comprehensive and detailed representation of the trajectories. To achieve these operations, we additionally introduce combined Attention and Mamba techniques for global information aggregation and state sequence modeling, leveraging their respective strengths. Extensive experiments on both the Argoverse 2 and nuScenes benchmarks demonstrate that our DeMo achieves state-of-the-art performance in motion forecasting.

## 1   Introduction

Motion forecasting [29, 58, 67] empowers self-driving vehicles to anticipate how surrounding agents will move and affect the ego car, providing references and conditions for the ego-action. It is critical for maintaining safety and dependability, enabling vehicles to comprehend the dynamics of driving environments and make calculated decisions. The challenges and complexities of this task arise from various factors, including unpredictable road conditions, varied movement patterns of traffic participants, and the necessity to simultaneously analyze the states of observed agents along with the road maps.

The research community has witnessed significant progress in the representation of driving scenes [18, 36, 44] and the paradigm of trajectory decoding [26, 38, 54, 77]. These methods have achieved substantial advancements in prediction accuracy, primarily following a certain pattern inspired from detection [4, 40], *i.e.*, the one-query-one-trajectory paradigm [38, 54, 59, 77]. This paradigm utilizes several queries to represent different estimated trajectories, as shown in Figure 1 (a), covering the possibilities of distinct motion intentions. Although effective, these approaches can only approximately provide a direction and collect surroundings to generate various trajectory waypoints in a one-shot fashion, overlooking the detailed relationships with scenes. The lack of concrete representation for trajectories and comprehensive spatiotemporal interactions with the surrounding

---

[*]Li Zhang (lizhangfd@fudan.edu.cn) is the corresponding author.

38th Conference on Neural Information Processing Systems (NeurIPS 2024).

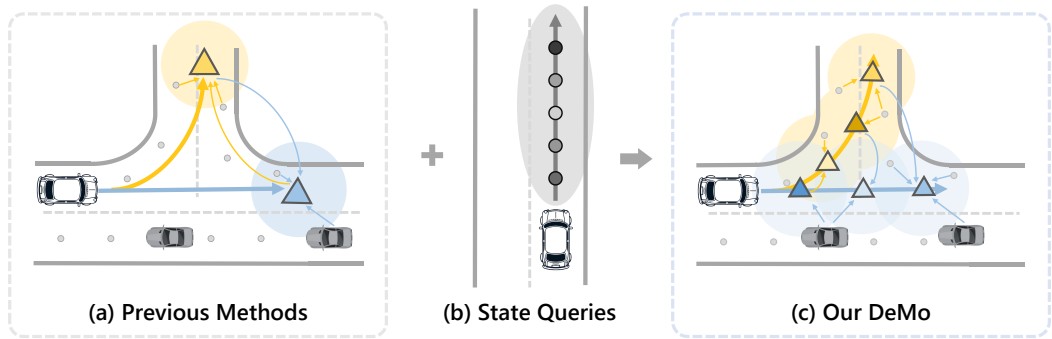



(a) Previous Methods      (b) State Queries      (c) Our DeMo



Figure 1: The primary distinction between previous methods and ours lies in the representation of future trajectories. Previous methods, as depicted in (a), use only one mode query for each trajectory. Our approach, illustrated in (c), utilizes decoupled mode queries, as shown in (a), and state queries, as shown in (b), to represent the multi-modal trajectories.

environment and among each other might lead to a decline in accuracy and consistency across varying time steps.

To solve this problem, we propose a novel framework dubbed **DeMo**, which provides a detailed representation of multi-modal trajectories. Specifically, we decouple forecasting queries into two types: besides the original motion mode queries to capture different directional intentions as shown in Figure 1 (a), we introduce the dynamic state queries for future trajectories to track the agent's dynamic states across various time steps, as shown in Figure 1 (b). This approach allows us to achieve a comprehensive query representation within our framework, as illustrated in Figure 1 (c). Mode queries and state queries are processed using the Mode Localization Module and the State Consistency Module, respectively. These modules enable the explicit interactions of queries with the surrounding environments and among each other, by which the directional accuracy and temporal consistency of future trajectories are significantly optimized. Subsequently, two types of queries are integrated by our Hybrid Coupling Module to achieve a comprehensive representation of future trajectories. Due to the sequential nature of trajectory states, Mamba is particularly well-suited for modeling the temporal consistency of dynamic states. Therefore, we utilize a combination of Attention and Mamba in our modules to effectively and efficiently aggregate global information and model state sequences, leveraging the strengths of both techniques.

Our contributions are summarized as follows: **(i)** We propose a motion forecasting framework that decouples multi-modal trajectory queries into mode queries and state queries to represent directional intentions and dynamic states, respectively. **(ii)** We design three modules based on integrated Attention and Mamba to process decoupled mode queries, state queries, and coupled mode and state queries. **(iii)** Extensive experiments on both the Argoverse 2 and nuScenes benchmarks demonstrate that DeMo achieves state-of-the-art performance.

## 2 Related work

**Motion forecasting.** In recent advancements in autonomous driving, it is critical to effectively predict the movements of relevant agents by accurately representing scene components. Traditional methods [5, 20, 48] transformed driving scenarios into image formats and used conventional convolutional networks for scene context encoding. However, these techniques often failed to sufficiently capture intricate structural details. This challenge has led to the adoption of vectorized scene representations [26, 60, 75, 79], exemplified by the introduction of VectorNet [18]. Additionally, graph-based structures are also widely utilized to represent the relationships between agents and their environments [14, 21, 30, 31, 36, 50, 69].

Existing methodologies have delved into a variety of frameworks to predict multi-modal future trajectories given the scene features. Initially, prediction techniques were centered on goal-oriented methods [26, 71] or employed probability heatmaps to sample trajectories [20, 21]. However, contemporary strategies, such as MTR [54] and QCNet [77], among others [41, 43, 44, 73], utilize Transformer [61] models to analyze relationships within the scene. Additionally, the introduction

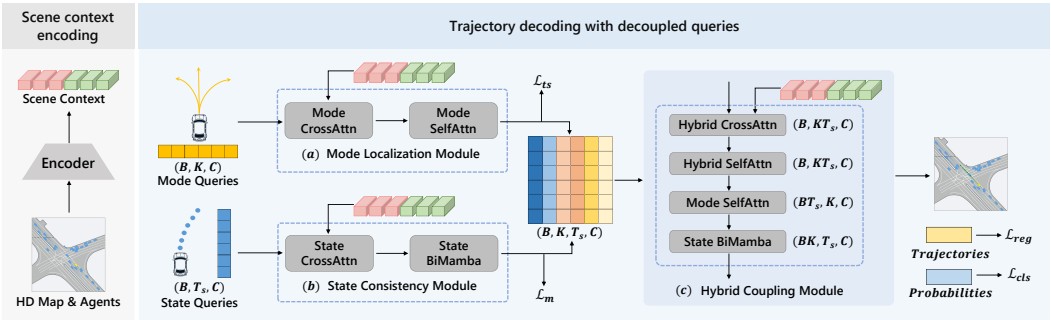

Figure 2: Overview of our DeMo framework: The HD maps and agents are first processed by the encoder to obtain the scene context. The decoding pipeline includes: (a) the Mode Localization Module, which processes mode queries by interacting with the scene context from the encoder and among themselves; (b) the State Consistency Module, which processes state queries; and (c) the Hybrid Coupling Module, which combines these queries to generate the final output. The feature dimension is illustrated using a single-agent setting, where $B$ represents the batch size.

of novel paradigms such as pre-training [7, 8, 34], historical prediction design [46, 59], GPT-style next-token prediction [49, 53], and post-refinement [9, 76] in some techniques has led to remarkable advancements in performance.

Furthermore, the advancements in multi-agent forecasting aim to enhance the applicability of predicted trajectories for various agents in real-world scenarios. Several approaches [22, 55, 79] follow an agent-centric model, where trajectories are forecasted individually for each agent, a process that might be slow. On the other hand, alternative approaches [44, 78] utilize a scene-centric model that allows for simultaneous forecasting across all agents, introducing an innovative approach to trajectory prediction.

Inspired by the progress in object detection and motivated by its significant success [4, 40], mainstream methods [38, 54, 59, 77] have adopted a one-query-one-trajectory paradigm to achieve high performance in motion forecasting benchmarks [6, 58, 67]. These methods use transformers to model the relationship between each trajectory query and its environment but lack detailed representation. We propose decoupled mode queries and state queries for a more detailed and comprehensive representation of multi-modal trajectories.

**State space models.** Originally developed for modeling dynamic systems with state variables in fields such as control theory, state space models (SSMs) have emerged as promising alternatives to Transformers [61] in sequence modeling, particularly due to their effectiveness in addressing attention complexity and capturing long-term dependencies. As SSMs have evolved [17, 25, 56], a new class termed Mamba [24], which incorporates selection mechanisms and hardware-aware architectures, has recently demonstrated significant promise in long-sequence modeling. Several studies have explored Mamba's substantial potential across a range of fields, including natural language processing [27, 37] and computer vision [28, 35, 74, 80]. Notably, in the vision domain, Mamba has demonstrated superior GPU efficiency and effectiveness compared to Transformers in tasks such as visual representation learning [80], video understanding [35], and human motion generation [74]. Building on these achievements, to the best of our knowledge, this is the first method to combine the strengths of Mamba with mainstream Transformer-based architecture to achieve impressive performance.

## 3 Methodology

In this section, we present DeMo, which utilizes decoupled mode queries and state queries for directional intentions and dynamic states to predict future trajectories. We also employ a hybrid architecture combining Attention and Mamba, along with two auxiliary losses for feature modeling.

### 3.1 Problem formulation

Given HD map and agents in the driving scenario, motion forecasting aims to predict the future trajectories for the interested agents. The HD map comprises several polylines of lanes or crossings, while agents are traffic participants like vehicles and pedestrians. To transform these elements into easily processable and learnable inputs, we utilize a popular vectorized representation following [18]. Specifically, the map $M \in \mathbb{R}^{N_m \times L \times C_m}$ is generated by dividing each line into several shorter segments, where $N_m$, $L$, and $C_m$ denote the number of map polylines, divided segments and feature channels, respectively. We represent the historical information of agents as $A \in \mathbb{R}^{N_a \times T_h \times C_a}$, where $N_a$, $T_h$, and $C_a$ are the number of agents, historical timestamps, and motion states (e.g., position, heading angle, velocity). Additionally, the future trajectories $A_f \in \mathbb{R}^{N_{aoi} \times T_f \times 2}$ for agents of interest are estimation objectives, with $N_{aoi}$, $T_f$ indicating the number of selected agents and the future timestamps, respectively.

### 3.2 Scene context encoding

Given the vectorized representations $A$ for agents and $M$ for HD map, we first employ individual encoders to process them separately. Specifically, we use a PointNet-based polyline encoder, as described in [8, 54, 55], to process the map representation $M$, generating the map features $F_m \in \mathbb{R}^{N_m \times C}$. For the agents $A$, we replace generic Transformer [61] or RNN with several Unidirectional Mamba [24] blocks, which are more efficient and effective for sequence encoding, to aggregate the historical trajectory features $F_a \in \mathbb{R}^{N_a \times C}$ up to the current time. Subsequently, the scene context features $F_s \in \mathbb{R}^{(N_a + N_m) \times C}$ are formed by concatenating them and further propagated to a Transformer encoder for intra-interaction learning. The overall process can be formulated as:

$$F_m = \text{PointNet}(M), \quad F_a = \text{UniMamba}(A), \quad F_s = \text{Transformer}(\text{Concat}(F_a, F_m)). \quad (1)$$

### 3.3 Trajectory decoding with decoupled queries

After obtaining the scene context features, we aim to decode multi-modal future trajectories for each interested agent based on our proposed decoupled queries. As illustrated in Figure 2, the decoder network comprises a State Consistency Module that enhances the consistency and accuracy of dynamic future state queries, a Mode Localization Module for learning distinct motion modes, and a Hybrid Coupling Module to integrate the decoupled queries and generate the final output. The detailed description of these components are provided in the following.

**Dynamic state consistency.** Considering the recurrence and causality of the future trajectories $A_f$, we propose to represent them as a series of dynamic states across various time steps, distinct yet interconnected. To preserve precise time information, the state queries $Q_s \in \mathbb{R}^{N_{aoi} \times T_s \times C}$ are initialized with an MLP module for real-time differences. It is notable that the steps $T_s$ can differ from $T_f$ to balance the effectiveness and efficiency, especially when predicting long-term future trajectories or a higher frequency of future trajectories. The State Consistency Module is then employed to enhance the consistency of the state queries and aggregate the specific scene context, which can be formulated as follows:

$$\begin{aligned}
Q_s &= \text{MLP}([t_1, t_2, \cdots, t_{T_s}]), \\
Q_s &= \text{MultiHeadAttn}(Q = Q_s, K = F_s, V = F_s), \\
Q_s &= \text{BiMamba}(Q_s).
\end{aligned} \quad (2)$$

Specifically, cross-attention is first applied to enable state queries to interact with the scene context, followed by a Mamba block to model sequence relationships with linear-time complexity. Simultaneously, to account for the influences of rear state queries on the front ones, we adopt the bidirectional Mamba [35, 80] for both forward and backward scanning. Additionally, a simple MLP module is utilized to decode the state queries $Q_s$ into a single future trajectory for explicit supervision of time consistency.

**Directional intention localization.** Mode queries $Q_{\mathrm{m}} \in \mathbb{R}^{N_{\mathrm{aoi}} \times K \times C}$ represent different motion modes, with each query responsible for decoding one of the $K$ trajectories. We utilize the Mode Localization Module to localize the potential directional intentions, as shown below:

$$
\begin{aligned}
Q_{\mathrm{m}} &= \mathrm{MultiHeadAttn}(\mathrm{Q} = Q_{\mathrm{m}}, \mathrm{K} = F_{\mathrm{s}}, \mathrm{V} = F_{\mathrm{s}}), \\
Q_{\mathrm{m}} &= \mathrm{MultiHeadAttn}(\mathrm{Q} = Q_{\mathrm{m}}, \mathrm{K} = Q_{\mathrm{m}}, \mathrm{V} = Q_{\mathrm{m}}).
\end{aligned}
\tag{3}
$$

For spatial motion learning, two Multi-Head Attention blocks are employed to enable interactions among mode queries and with the scene context. Additionally, we also employ simple MLPs to decode the future trajectories and probabilities. Similarly, we introduce another auxiliary supervision to endow mode queries with distinct motion intentions.

**Hybrid query coupling.** To incorporate dynamic states and directional intentions, we simply add $Q_{\mathrm{m}}$ and $Q_{\mathrm{s}}$ together to form the hybrid spatiotemporal queries $Q_{\mathrm{h}} \in \mathbb{R}^{N_{\mathrm{aoi}} \times K \times T_{\mathrm{s}} \times C}$. Then, the Hybrid Coupling Module is utilized to further process $Q_{\mathrm{h}}$ and yield a comprehensive representation for future trajectories, as formulated below:

$$
\begin{aligned}
Q_{\mathrm{h}} &= \mathrm{MultiHeadAttn}(\mathrm{Q} = Q_{\mathrm{h}}, \mathrm{K} = F_{\mathrm{s}}, \mathrm{V} = F_{\mathrm{s}}), \\
Q_{\mathrm{h}} &= \mathrm{HybridMultiHeadAttn}(\mathrm{Q} = Q_{\mathrm{h}}, \mathrm{K} = Q_{\mathrm{h}}, \mathrm{V} = Q_{\mathrm{h}}), \\
Q_{\mathrm{h}} &= \mathrm{ModeMultiHeadAttn}(\mathrm{Q} = Q_{\mathrm{h}}, \mathrm{K} = Q_{\mathrm{h}}, \mathrm{V} = Q_{\mathrm{h}}), \\
Q_{\mathrm{h}} &= \mathrm{BiMamba}(Q_{\mathrm{h}}).
\end{aligned}
\tag{4}
$$

Besides the Attention and Mamba modules for interaction with the scene context, among modes, and across time states, we additionally introduce a hybrid self-attention layer, which connects queries across both time and modes, boosting the diversity of predicted trajectories. The change in feature dimensions in this module is shown in Figure 2. The final predictions are generated by decoding the output $Q_{\mathrm{h}}$ into trajectory positions and probabilities with MLPs.

### 3.4 Training losses

DeMo is trained with three component losses in an end-to-end manner. Primarily, the regression loss $\mathcal{L}_{\mathrm{reg}}$ and the classification loss $\mathcal{L}_{\mathrm{cls}}$ are employed to supervise the accuracy of predicted trajectories and their associated probability scores. Additionally, we introduce two auxiliary losses, $\mathcal{L}_{\mathrm{ts}}$ and $\mathcal{L}_{\mathrm{m}}$, for intermediate features of time states and motion modes, respectively. The former enhances the coherence and causality of dynamic states across various time steps, while the latter endows mode with distinct directional intentions. The overall loss $\mathcal{L}$ is a combination of these individual losses with equal weights, formulated as follows:

$$
\mathcal{L} = \mathcal{L}_{\mathrm{reg}} + \mathcal{L}_{\mathrm{cls}} + \mathcal{L}_{\mathrm{ts}} + \mathcal{L}_{\mathrm{m}}.
\tag{5}
$$

We adopt the cross-entropy loss for probability score classification and the smooth-L1 loss for trajectory regression tasks. The winner-take-all strategy is employed, optimizing only the best prediction with minimal average prediction error to the ground truth.

## 4 Experiments

### 4.1 Experimental settings

**Datasets.** We evaluate our method's performance using the Argoverse 2 [67] and nuScenes [3] motion forecasting datasets. The Argoverse 2 dataset comprises 250,000 scenarios with a sampling frequency of 10 Hz, each featuring a 5s historical trajectory length and predicting a 6s future ones. The nuScenes dataset contains 1,000 scenes at 2 Hz, predicting the next 6s trajectories with the past 2s history.

Table 1: Performance comparison on *Argoverse 2 single-agent test set* in the official leaderboard. For each metric, the best result is in **bold** and the second best result is underlined. The upper part features a single model, while the lower part employs model ensembling as a trick.

| Method | $minFDE_1$ | $minADE_1$ | $minFDE_6$ | $minADE_6$ | $MR_6$ | $b\text{-}minFDE_6$ |
|---|---|---|---|---|---|---|
| FRM [47] | 5.93 | 2.37 | 1.81 | 0.89 | 0.29 | 2.47 |
| HDGT [31] | 5.37 | 2.08 | 1.60 | 0.84 | 0.21 | 2.24 |
| SIMPL [72] | 5.50 | 2.03 | 1.43 | 0.72 | 0.19 | 2.05 |
| THOMAS [22] | 4.71 | 1.95 | 1.51 | 0.88 | 0.20 | 2.16 |
| GoRela [11] | 4.62 | 1.82 | 1.48 | 0.76 | 0.22 | 2.01 |
| MTR[54] | 4.39 | 1.74 | 1.44 | 0.73 | 0.15 | 1.98 |
| HPTR [73] | 4.61 | 1.84 | 1.43 | 0.73 | 0.19 | 2.03 |
| GANet [64] | 4.48 | 1.77 | 1.34 | 0.72 | 0.17 | 1.96 |
| ProphNet [65] | 4.74 | 1.80 | 1.33 | 0.68 | 0.18 | 1.88 |
| QCNet [77] | 4.30 | 1.69 | 1.29 | 0.65 | 0.16 | 1.91 |
| SmartRefine [76] | 4.17 | 1.65 | 1.23 | 0.63 | 0.15 | 1.86 |
| **DeMo (Ours)** | **3.74** | **1.49** | **1.17** | **0.61** | **0.13** | **1.84** |
| QML [57] | 4.98 | 1.84 | 1.39 | 0.69 | 0.19 | 1.95 |
| TENET [66] | 4.69 | 1.84 | 1.38 | 0.70 | 0.19 | 1.90 |
| MacFormer [15] | 4.69 | 1.84 | 1.38 | 0.70 | 0.19 | 1.90 |
| BANet [70] | 4.61 | 1.79 | 1.36 | 0.71 | 0.19 | 1.92 |
| Gnet [19] | 4.40 | 1.72 | 1.34 | 0.69 | 0.18 | 1.90 |
| Forecast-MAE [8] | 4.15 | 1.66 | 1.34 | 0.69 | 0.17 | 1.91 |
| QCNet [77] | 3.96 | 1.56 | 1.19 | 0.62 | 0.14 | 1.78 |
| **DeMo (Ours)** | **3.70** | **1.49** | **1.11** | **0.60** | **0.12** | **1.73** |

**Evaluation metrics.** We adopt the common metrics: $minADE$, $minFDE$, $MR$, and $b\text{-}minFDE$. For multi-agent scenarios, we use $avgMinADE$, $avgMinFDE$, and $actorMR$. The Argoverse 2 dataset is evaluated across six prediction modes, while nuScenes is evaluated across ten prediction modes. We typically follow the evaluation metrics from the official leaderboard, setting $K$ to 1 and 6 for the Argoverse 2 dataset, and $K$ to 5 and 10 for the nuScenes dataset.

**Implementation details.** Our models are trained for 60 epochs using the AdamW [42] optimizer, with a batch size of 16 per GPU. The training is conducted end-to-end with a learning rate of 0.003 and a weight decay of 0.01. We adopt an agent-centric coordinate system and sample scene elements within a 150-meter radius of the agents of interest. All experiments are conducted on 8 NVIDIA GeForce RTX 3090 GPUs. Additional details and further experiments are provided in Appendix A and Appendix B.

## 4.2 Comparison with state of the art

We first compare our method, DeMo with several models on the Argoverse 2 [67] motion forecasting benchmark for the single-agent setting as demonstrated in Table 1. To ensure a comprehensive and fair comparison, we separately evaluate the performance of different methods with and without the model ensembling technique. It is shown that DeMo has significantly outperformed all previous approaches, including the state-of-the-art model QCNet [77] and its post-refinement enhancement SmartRefine [76]. Concretely, our method distinctly surpasses other methods across all metrics, particularly in terms of $minFDE_1$ and $minADE_1$, where it demonstrates performance improvements of 13.02% and 11.83% relative to QCNet, respectively. After using ensembling techniques similar to other entries, DeMo surpasses all methods on all metrics by a large margin. Then, we compare the performance of DeMo on the nuScenes [3] motion forecasting benchmark, with the results of the test split presented in Table 2. Our method is also superior to others over all metrics except $minADE_5$.

Table 2: Performance comparison on *nuScenes test set* in the official leaderboard. The "-" symbol means the corresponding metric is unknown.

| Method | $minFDE_1$ | $minADE_5$ | $minADE_{10}$ | $MR_5$ | $MR_{10}$ |
|---|---|---|---|---|---|
| Trajectron++ [51] | 9.52 | 1.88 | 1.51 | 0.70 | 0.57 |
| LaPred [33] | 8.37 | 1.47 | 1.12 | 0.53 | 0.46 |
| P2T [13] | 10.50 | 1.45 | 1.16 | 0.64 | 0.46 |
| GOHOME [21] | 6.99 | 1.42 | 1.15 | 0.57 | 0.47 |
| CASPNet [52] | - | 1.41 | 1.19 | 0.60 | 0.43 |
| Autobot [23] | 8.19 | 1.37 | 1.03 | 0.62 | 0.44 |
| THOMAS [22] | 6.71 | 1.33 | 1.04 | 0.55 | 0.42 |
| PGP [14] | 7.17 | 1.27 | 0.94 | 0.52 | **0.34** |
| LAformer [39] | 6.95 | **1.19** | 1.19 | 0.48 | 0.48 |
| **DeMo (Ours)** | **6.60** | 1.22 | **0.89** | **0.43** | **0.34** |

Table 3: Performance comparison on *Argoverse 2 multi-agent test set* in the official leaderboard.

| Method | $avgMinFDE_1$ | $avgMinADE_1$ | $avgMinFDE_6$ | $avgMinADE_6$ | $actorMR_6$ |
|---|---|---|---|---|---|
| FJMP [50] | 4.00 | 1.52 | 1.89 | 0.81 | 0.23 |
| Forecast-MAE [8] | 3.33 | 1.30 | 1.55 | 0.69 | 0.19 |
| FFINet [32] | 3.18 | 1.24 | 1.77 | 0.77 | 0.24 |
| Gnet [19] | 3.05 | 1.23 | 1.46 | 0.69 | 0.19 |
| **DeMo (Ours)** | **2.78** | **1.12** | **1.24** | **0.58** | **0.16** |

## 4.3 Multi-agent quantitative results

In multi-agent environments, it is essential for predictors to simultaneously forecast the future paths of all relevant agents to comprehensively understand the driving situation. To validate the efficacy of our model, DeMo, we conduct tests on the Argoverse 2 multi-agent dataset [67]. The results, presented in Table 3, show that despite lacking the specialized multi-agent forecasting features found in models such as [44, 55, 78], our model surpasses recent advancements across all evaluated metrics due to our novel designs.

## 4.4 Ablation study

**Effects of components.** Table 4 demonstrates the effectiveness of each component in our method. We show the baseline in the first row, which is similar to previous methods [54, 77] and utilizes mode queries to generate multi-modal future trajectories. Then, we directly adopt state queries in the second row (ID-2) to decode the trajectories. A performance decline is observed due to the surplus queries, which impose a burden on the model and make it difficult to distinguish the meanings of different types. In the third row (ID-3), we introduce two auxiliary losses, resulting in a slight improvement compared to the first row. Although the model can identify what each query represents, it demonstrates only moderate performance due to the limited information. In the fourth row (ID-4), we incorporate the three aggregation modules in Figure 2 but remove auxiliary losses, leading to significant performance enhancements. Finally, in the fifth row (ID-5), our DeMo integrates all these techniques and achieves outstanding performance.

**Effects of state sequence modeling with Mamba.** Mamba excels at sequence modeling, so we utilize Bidirectional Mamba [35, 80] to enhance the consistency of states across different time steps. To demonstrate its effectiveness, we compare Bidirectional Mamba with several other modules, including Unidirectional Mamba [24], Attention, Conv1d, and GRU [10]. As illustrated in the left part of Table 5, our Bidirectional Mamba configuration outperforms the others due to its specialized design for sequence modeling, compared to Attention, and its capability to perform both forward and backward scans, unlike Unidirectional Mamba.

Table 4: Ablation study on the core components of DeMo on the *Argoverse 2 single-agent validation set*. "Decpl. Query" indicates decoupled query paradigm. "Agg. Module" indicates three aggregation modules. "Aux. Loss" indicates two auxiliary losses.

| ID | State Query | Decpl. Query | Agg. Module | Aux. Loss | $minFDE_1$ | $minADE_1$ | $minFDE_6$ | $minADE_6$ | $MR_6$ | $b\text{-}minFDE_6$ |
|----|-----|-----|-----|-----|-----|-----|-----|-----|-----|-----|
| 1 | | | | | 4.489 | 1.792 | 1.414 | 0.750 | 0.184 | 2.067 |
| 2 | ✓ | | | | 4.494 | 1.800 | 1.505 | 0.777 | 0.208 | 2.138 |
| 3 | ✓ | ✓ | | ✓ | 4.385 | 1.746 | 1.405 | 0.761 | 0.180 | 2.051 |
| 4 | ✓ | ✓ | ✓ | | 4.247 | 1.695 | 1.319 | 0.687 | 0.166 | 1.961 |
| 5 | ✓ | ✓ | ✓ | ✓ | **3.917** | **1.609** | **1.268** | **0.674** | **0.152** | **1.918** |

Table 5: Ablation study on (a) (left) the sequence modeling choices and (b) (right) the effects of aggregation modules and auxiliary losses. For (a), "Uni-MB" and "Bi-MB" represent Unidirectional Mamba and Bidirectional Mamba. For (b), "H.C." indicates Hybrid Coupling Module. "S.C." indicates State Consistency Module. "M.L." indicates Mode Localization Module.

| | $minFDE_6$ | $minADE_6$ | $MR_6$ | | $minFDE_6$ | $minADE_6$ | $MR_6$ |
|----|-----|-----|-----|----|-----|-----|-----|
| None | 1.307 | 0.692 | 0.161 | w/o $\mathcal{L}_{ts}$ | 1.290 | 0.715 | 0.161 |
| GRU | 1.842 | 0.923 | 0.274 | w/o $\mathcal{L}_m$ | 1.289 | 0.687 | 0.159 |
| Conv1d | 1.304 | 0.693 | 0.161 | w/o H.C. | 1.324 | 0.704 | 0.164 |
| Attn | 1.289 | 0.687 | 0.159 | w/o S.C. | 1.317 | 0.697 | 0.162 |
| Uni-MB | 1.288 | 0.690 | 0.156 | w/o M.L. | 1.297 | 0.693 | 0.158 |
| Bi-MB | **1.268** | **0.674** | **0.152** | All | **1.268** | **0.674** | **0.152** |

**Effects of auxiliary losses and aggregation modules.** We conduct an ablation study to assess the impacts of auxiliary losses and aggregation modules. As illustrated in the right part of Table 5, removing any of these losses or modules leads to a performance decline in the model. Notably, the aggregation modules have a greater impact than the auxiliary losses. This is attributed to the critical role of learning information from the scene context and from each other, which is essential for decoupling queries to represent distinct meanings.

**Effects of state queries.** We conduct an ablation study on the number of state queries, as shown in the left part of Table 6. In our default setting, we use 60 state queries to represent the future states at 60 timestamps. As we gradually reduce the number of state queries, we observe a performance decline due to the increasing ambiguity of the state query meanings.

**Effects of the depth of Attention and Mamba blocks.** A suitable depth configuration of Attention and Mamba units is crucial for achieving an optimal balance between efficiency and performance. As depicted in the right part of Table 6, we conduct an ablation study focusing on the layer depth. It is observed that the best results are obtained with Attention units at a depth of three and Mamba units at a depth of two.

Table 6: Ablation study on (a) (left) state queries and (b) (right) the depth of Attention and Mamba layers.

| Queries | $minFDE_6$ | $minADE_6$ | $MR_6$ | Attn | M.B. | $minFDE_6$ | $minADE_6$ | $MR_6$ |
|----|-----|-----|-----|----|----|-----|-----|-----|
| 10 | 1.312 | 0.704 | 0.160 | 1 | 1 | 1.309 | 0.708 | 0.160 |
| 20 | 1.294 | 0.688 | 0.157 | 2 | 2 | 1.288 | 0.691 | 0.157 |
| 30 | 1.290 | 0.692 | 0.155 | 3 | | **1.268** | **0.674** | **0.152** |
| 60 | **1.268** | **0.674** | **0.152** | | 3 | 1.276 | 0.675 | 0.154 |

**Effects of the depth of Mamba blocks in the encoder.** We add ablation studies on the Mamba for encoding agent historical information in the encoder of our DeMo. As shown in Table 7, the left part shows different modules for encoding the historical information of agents. Our goal is to aggregate historical information up to the present time, making Unidirectional Mamba the most suitable choice.

The right part presents an ablation study concerning the number of Mamba blocks, indicating that three layers yield the optimal performance.

Table 7: Ablation study on (a) (left) sequence modeling choices and (b) (right) depth of Mamba blocks in agent historical information encoding.

| | $minFDE_6$ | $minADE_6$ | $MR_6$ | Num | $minFDE_6$ | $minADE_6$ | $MR_6$ |
|---|---|---|---|---|---|---|---|
| GRU | 1.344 | 0.726 | 0.170 | 1 | 1.312 | 0.701 | 0.162 |
| Bi-MB | 1.280 | 0.684 | 0.154 | 2 | 1.283 | 0.681 | 0.155 |
| Uni-MB | **1.268** | **0.674** | **0.152** | 3 | **1.268** | **0.674** | **0.152** |

## 4.5 An analysis to improve the measurement of query decoupling

We measure the outputs of state queries and mode queries with $minADE$ and $minFDE$, as shown in Table 8. We can see that the $minADE_1$ and $minFDE_1$ of the trajectories from state query outputs are better than those from mode query outputs. This means state dynamics are encoded in state queries. Additionally, there are six output trajectories from mode queries, indicating that directional information is predominantly stored in mode queries. The final outputs take advantage of the strengths of both.

Table 8: An analysis to improve the measurement of query decoupling.

| | $minFDE_1$ | $minADE_1$ | $minFDE_6$ | $minADE_6$ |
|---|---|---|---|---|
| state query outputs | 3.84 | 1.52 | - | - |
| mode query outputs | 4.12 | 1.63 | 1.31 | 0.67 |
| final outputs | 3.93 | 1.54 | 1.24 | 0.64 |

## 4.6 Efficiency analysis and qualitative results

Balancing performance, inference speed, and model size is important for the model deployment. We compare our DeMo with two recent, representative models: the state-of-the-art QCNet [77] and its enhancement through post-refinement, SmartRefine [76]. The size of our model is 5.9M, in contrast to 7.7M for QCNet and 8.0M for SmartRefine. Despite its smaller size, our model demonstrates superior performance by a significant margin, as detailed in Table 1. As for inference speed, we compare DeMo with QCNet, both end-to-end methods. Measurements are conducted on the Argoverse 2 single-agent validation set using an NVIDIA GeForce RTX 3090 GPU, with a maintained batch size of one. The average inference speed of DeMo is only 38ms, which is approximately 2.5 times faster than QCNet's 94ms. This demonstrates that our method is not only superior to QCNet but also more efficient.

In Figure 3, we present qualitative results of our network. The results of the baseline model, which lacks the decoupled query paradigm, are shown in panel (a), while the results of our DeMo are shown in panel (b). From the first two rows, it is evident that by explicitly optimizing the dynamic states of future trajectories, our model predicts trajectories that are more accurate and closer to the ground truth. From the third row, it is apparent that our model can better capture potential directional intentions. Additional qualitative results and failure cases are detailed in Appendix D and E.

## 4.7 Computational cost compared to other methods

We provide a comparison of computational cost with resent representative methods in Table 9. The experiments are conducted on Argoverse 2 [67] dataset using 8 NVIDIA GeForce RTX 3090 GPUs.

Table 9: Computational cost compared to other methods.

| Method | FLOPs | Training time | Memory | Parameter | Batch size |
|---|---|---|---|---|---|
| SIMPL [72] | 19.7 GFLOPs | 8h | 14G | 1.9M | 16 |
| QCNet [77] | 53.4 GFLOPs | 45h | 16G | 7.7M | 4 |
| **DeMo (Ours)** | 22.8 GFLOPs | 9h | 12G | 5.9M | 16 |

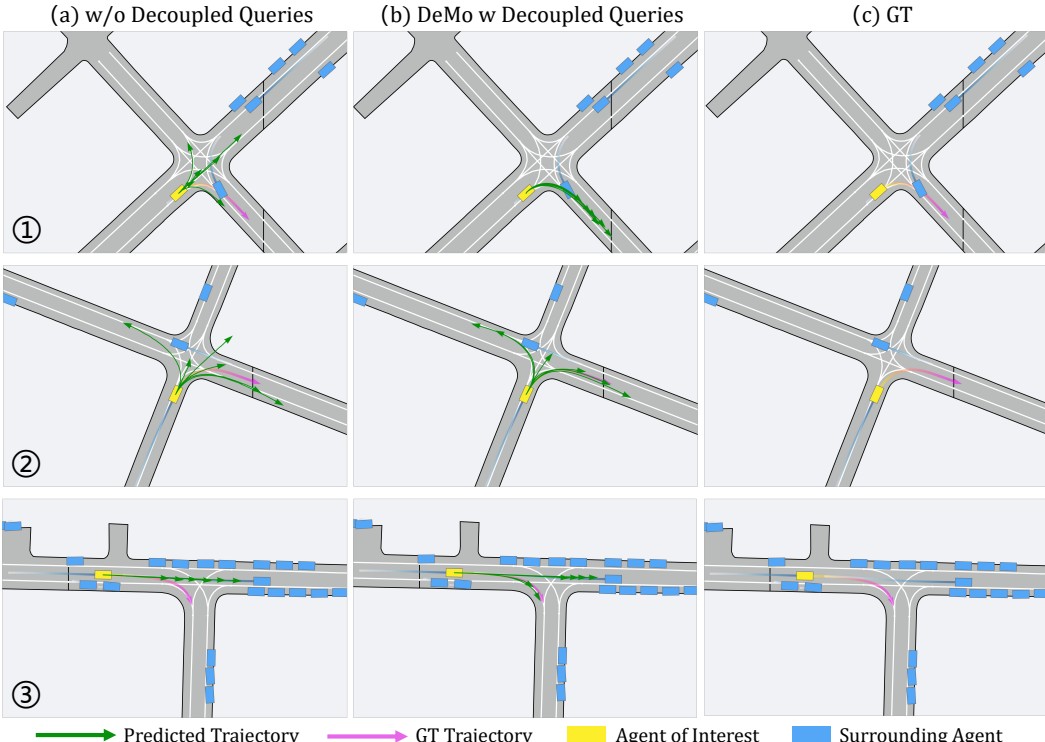

Figure 3: Qualitative results on the Argoverse 2 single-agent validation set. Panel (a) illustrates the results of the baseline model without decoupled queries; Panel (b) illustrates the results of our DeMo, which employs decoupled queries; and Panel (c) represents the ground truth.

## 5 Conclusion

In this paper, we introduce DeMo, which redefines the motion forecasting task by decoupling it into expressions of directional intentions and dynamic states. We utilize state queries to model various states across different time, and mode queries to capture the agent's motion intentions. Our approach incorporates three aggregation modules, combining Attention and Mamba for effective modeling. Comprehensive experiments, covering both single-agent and multi-agent scenarios, indicate that DeMo outperforms the current state-of-the-art methods. This highlights its potential as a promising approach for achieving safe and reliable motion forecasting in the rapidly advancing field of autonomous driving.

**Limitations and future work.** The proposed framework adopts a decoupled query paradigm, which may lead to heavier models due to the need to predict longer trajectories. Our current model design does not sufficiently take model efficiency into account. In the future, we plan to use sparse states for modeling trajectories, thereby making the framework more deployment-friendly.

## Acknowledgments

This work was supported in part by National Natural Science Foundation of China (Grant No. 62106050 and 62376060), Natural Science Foundation of Shanghai (Grant No. 22ZR1407500).

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

# Appendix

## A   Experimental settings

### A.1   Baseline model

Based on the streaming nature of driving data, our baseline model employs a lightweight transformer-based architecture capable of historical prediction [45, 59]. This model predicts future trajectories at intervals of 3s, 4s, and 5s for the Argoverse 2 dataset, and at 1s, 1.5s, and 2s for the Argoverse 1 dataset. The final prediction is the ultimate outcome of these trajectories. Features from the encoder and decoder in the historical prediction phase are aggregated with those from the encoder and decoder in the subsequent prediction phase. The aggregation module comprises two cross-attention layers. The historical prediction component is not used for the nuScenes dataset due to the limited history steps. In ablation studies, we remove the historical prediction component from the baseline model to make the experiments more efficient.

### A.2   Single-agent evaluation metrics

We employ established metrics to evaluate our models, including minimum Average Displacement Error ($minADE_k$), minimum Final Displacement Error ($minFDE_k$), Miss Rate ($MR_k$), and Brier minimum Final Displacement Error ($b\text{-}minFDE_k$). The $minADE_k$ metric calculates the $L_2$ distance between the ground-truth trajectory and the best $K$ predicted trajectories, averaged over all future time steps. The $minFDE_k$ metric measures the discrepancy between the endpoints of the predicted trajectories and the ground truth. The $MR_k$ metric represents the proportion of scenes where $minFDE_k$ exceeds 2 meters. To provide a more nuanced evaluation of uncertainty, $b\text{-}minFDE_k$ incorporates $(1-\pi)^2$ into the final displacement error, where $\pi$ indicates the probability score assigned by the model to the best-predicted trajectory.

### A.3   Multi-agent evaluation metrics

Following the official settings for the Argoverse 2 multi-agent test set, we use metrics like Average Minimum Final Displacement Error ($avgMinFDE$), Average Minimum Average Displacement Error ($avgMinADE$), and Actor Miss Rate ($actorMR$). The $avgMinFDE$ metric calculates the average of the lowest Final Displacement Errors (FDEs) for all scored actors in a scenario, reflecting prediction accuracy. Similarly, $avgMinADE$ is the average of the lowest Average Displacement Errors (ADEs) for all scored actors, showing overall movement accuracy. The $actorMR$ measures the proportion of actors missed across the evaluation set, as previously described.

### A.4   More implementation details

In addition to the details in Section 4.1, we set the dropout rates at 0.2 for single-agent settings and 0.1 for multi-agent settings. We employ a cosine learning rate schedule with a warm-up phase of 10 epochs. For normalization, we use nn.LayerNorm, and for activation, we use nn.GELU. No data augmentation techniques are used. For details on the number of layers in each component, please refer to Table 10.

### A.5   A precise formulation of the auxiliary losses

We use an MLP to decode state queries into a single future trajectory $Y_f$ and calculate the loss with ground truth $Y_{gt}$ to obtain $\mathcal{L}_{ts}$:

$$\mathcal{L}_{ts} = \text{SmoothL1}(Y_f, Y_{gt}). \tag{6}$$

We use MLPs to decode the future trajectories $Y_f$ and probabilities $P_f$. So $\mathcal{L}_m$ is shown below:

$$\begin{aligned} Y_{best}, P_{best} &= \text{SelectBest}(Y_f, Y_{gt}), \\ \mathcal{L}_m &= \text{SmoothL1}(Y_{best}, Y_{gt}) + \text{CrossEntropy}(P_f, P_{best}). \end{aligned} \tag{7}$$

Table 10: Number of layers in each component.

| Enc/Dec | Name | Num-AV1&AV2 | Num-nuScenes |
|---|---|---|---|
| Enc | Agent Encoding Mamba | 4 | 2 |
| | Scene Context Transformer | 5 | 4 |
| Dec | State Consistency Module Attention | 2 | 2 |
| | State Consistency Module Mamba | 2 | 2 |
| | Mode Localization Module Attention | 3 | 2 |
| | Hybrid Coupling Module Attention | 3 | 2 |
| | Hybrid Coupling Module Mamba | 2 | 2 |

## B    More experiments

### B.1    Performance comparison on the Argoverse 1 dataset

To fully demonstrate the effectiveness of our DeMo, we compare it with several recent models on the Argoverse 1 [6] dataset. The results from the validation split are shown in Table 11, indicating that our model achieves impressive performance.

Table 11: Performance comparison on *Argoverse 1 validation set*.

| Method | $minADE_6$ | $minFDE_6$ | $MR_6$ |
|---|---|---|---|
| LTP [62] | 0.78 | 1.07 | - |
| LaneRCNN [69] | 0.77 | 1.19 | 0.08 |
| TPCN [68] | 0.73 | 1.15 | 0.11 |
| DenseTNT [26] | 0.73 | 1.05 | 0.10 |
| TNT [75] | 0.73 | 1.29 | 0.09 |
| mmTransformer [41] | 0.71 | 1.15 | 0.11 |
| LaneGCN [36] | 0.71 | 1.08 | - |
| SSL-Lanes [2] | 0.70 | 1.01 | 0.09 |
| PAGA [12] | 0.69 | 1.02 | - |
| DSP [71] | 0.69 | 0.98 | 0.09 |
| FRM [47] | 0.68 | 0.99 | - |
| ADAPT [1] | 0.67 | 0.95 | 0.08 |
| SIMPL [72] | 0.66 | 0.95 | 0.08 |
| HiVT [79] | 0.66 | 0.96 | 0.09 |
| R-Pred [9] | 0.66 | 0.95 | 0.09 |
| HPNet [59] | 0.64 | **0.87** | **0.07** |
| **DeMo (Ours)** | **0.59** | 0.90 | **0.07** |

### B.2    Performance on the Argoverse 2 leaderboard

We provide a performance comparison of the top methods on the Argoverse 2 leaderboard as of September 2024. The results for the single-agent setting are shown in Table 12, and the results for the multi-agent setting are shown in Table 13.

### B.3    Results on Waymo open motion dataset

We provide results on WOMD [58] in Table 14 using the settings in UniTraj [16], as shown below. The results of other methods are also from UniTraj.

### B.4    Model ensembling

In our approach, we use model ensembling, an essential technique to enhance the accuracy of final predictions. We train six sub-models with various random seeds and training epochs, resulting in 36

Table 12: Performance comparison of the single-agent setting on the Argoverse 2 dataset in the official leaderboard. Unreleased works are marked with the symbol "*".

| Method | $b\text{-}minFDE_6$ | Rank |
|---|---|---|
| LOF [63] | 1.63 | 1 |
| iDLab-SEPT++ (SEPT++) * | 1.65 | 2 |
| EACON (JMaC) * | 1.67 | 3 |
| PolarMotion_E (PolarMotion) * | 1.71 | 4 |
| **DeMo (Ours)** | 1.73 | 5 |
| iDLab-SEPT (SEPT) [34] | 1.74 | 6 |
| xPnC (X-MotionFormer) * | 1.74 | 7 |
| GACRND-XLAB (XPredFormer) * | 1.76 | 8 |

Table 13: Performance comparison of the multi-agent setting on the Argoverse 2 dataset in the official leaderboard. Unreleased works are marked with the symbol "*".

| Method | $avgBrierMinFDE_6$ | Rank |
|---|---|---|
| EACON (JMaC) * | 1.62 | 1 |
| QCNet-AV2 (QCNeXt) [78] | 1.65 | 2 |
| Lite-QCNet * | 1.67 | 3 |
| LOF [63] | 1.68 | 4 |
| iDLab-SEPT [34] | 1.80 | 5 |
| **DeMo (Ours)** | 1.93 | 6 |
| FAW-Prediction * | 1.93 | 7 |
| berste (OGD_test2) * | 1.95 | 8 |

Table 14: Results on WOMD.

| Method | $minFDE_6$ | $minADE_6$ |
|---|---|---|
| MTR | 1.78 | 0.78 |
| Wayformer | 1.46 | 0.65 |
| AutoBot | 1.65 | 0.73 |
| **DeMo (Ours)** | 1.59 | 0.75 |

predicted future trajectories for each agent. We then apply k-means clustering with six cluster centers to process these trajectories. For each cluster group, we compute the average trajectory within the group to determine the final trajectories. We present the results, both with and without the model ensembling technique, on the Argoverse 2 single-agent test set in Table 1.

## C  Mamba introduction

Mamba is inspired by a continuous system that maps a 1-D function or sequence $x(t) \in \mathbb{R}$ to $y(t) \in \mathbb{R}$, utilizing a hidden state $h(t) \in \mathbb{R}^N$. In this system, $\mathbf{A} \in \mathbb{R}^{N \times N}$ serves as the evolution parameter, while $\mathbf{B} \in \mathbb{R}^{N \times 1}$ and $\mathbf{C} \in \mathbb{R}^{1 \times N}$ act as the projection parameters.

$$
\begin{aligned}
h'(t) &= \mathbf{A}h(t) + \mathbf{B}x(t), \\
y(t) &= \mathbf{C}h(t).
\end{aligned}
\tag{8}
$$

Mamba represents the discrete version of a continuous system and includes a timescale parameter $\mathbf{\Delta}$ to convert the continuous parameters $\mathbf{A}$ and $\mathbf{B}$ into discrete counterparts $\overline{\mathbf{A}}$ and $\overline{\mathbf{B}}$. The most commonly used method for this transformation is zero-order hold (ZOH). After discretizing $\overline{\mathbf{A}}$ and $\overline{\mathbf{B}}$, the discretized form of Equation (8) using a step size of $\mathbf{\Delta}$ can be reformulated as:

$$\overline{\mathbf{A}} = \exp\left(\mathbf{\Delta}\mathbf{A}\right),$$
$$\overline{\mathbf{B}} = (\mathbf{\Delta}\mathbf{A})^{-1}(\exp\left(\mathbf{\Delta}\mathbf{A}\right) - \mathbf{I}) \cdot \mathbf{\Delta}\mathbf{B}.$$
$$h_t = \overline{\mathbf{A}}h_{t-1} + \overline{\mathbf{B}}x_t,$$
$$y_t = \mathbf{C}h_t.$$

(9)

Finally, the models compute the output through a global convolution that utilizes a structured convolutional kernel $\overline{\mathbf{K}} \in \mathbb{R}^M$, where $M$ represents the length of the input sequence $\mathbf{x}$.

$$\overline{\mathbf{K}} = (\mathbf{C}\overline{\mathbf{B}}, \mathbf{C}\overline{\mathbf{A}}\overline{\mathbf{B}}, \dots, \mathbf{C}\overline{\mathbf{A}}^{M-1}\overline{\mathbf{B}}),$$
$$\mathbf{y} = \mathbf{x} * \overline{\mathbf{K}},$$

(10)

## D More qualitative results

We provide more qualitative results of our DeMo in Figure 5.

## E Failure cases

Although our DeMo has demonstrated exceptional performance on motion forecasting benchmarks, it is not without its failures. We analyze these typical cases and present qualitative results to give readers insight into the scenarios where our model might underperform. This analysis is intended to support future efforts in developing an algorithm that is both more robust and powerful, as illustrated in Figure 4. In the first row, the vehicle will turn into an alley, reflecting a kind of subjective driving behavior. However, the model predicts that the vehicle will just keep going straight. To improve predictions in cases like this, we could enhance how the model interacts with additional information about what the vehicle intends to do, such as adding visual cues, such as turn signals. In the second row, the agent needs to navigate through a complex intersection to reach one of the roads; however, the model fails to accurately predict this driving behavior. This inaccuracy may be caused by a lack of comprehensive understanding of the complex map topology and the unbalanced distribution of driving data, addressing the issue of data balance is necessary to solve this problem.

(a) Our Model Output                    (b) GT

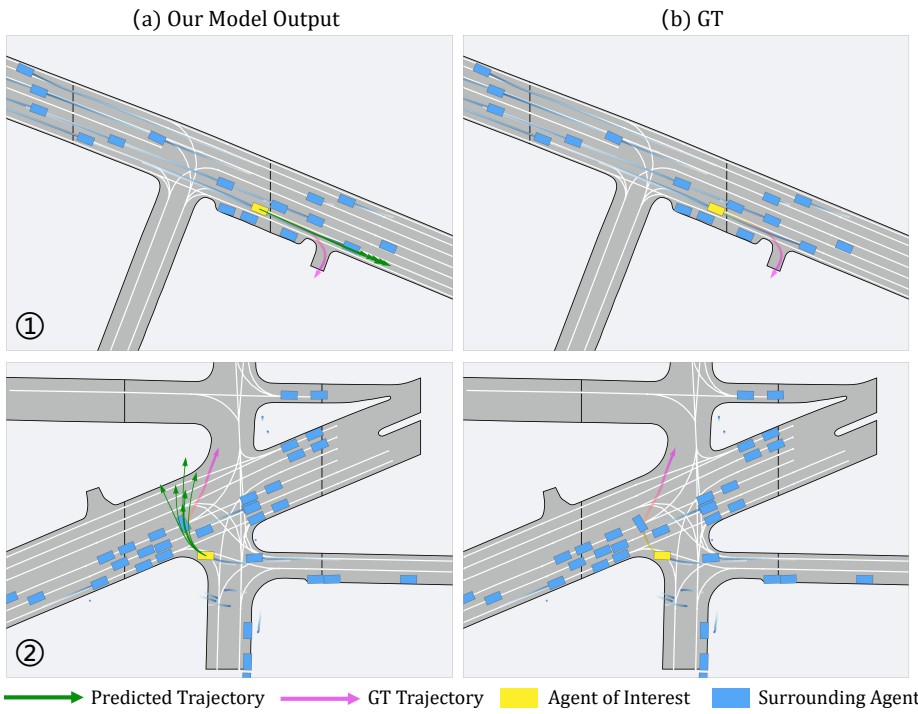

Predicted Trajectory — GT Trajectory — Agent of Interest — Surrounding Agent

Figure 4: Failure cases.

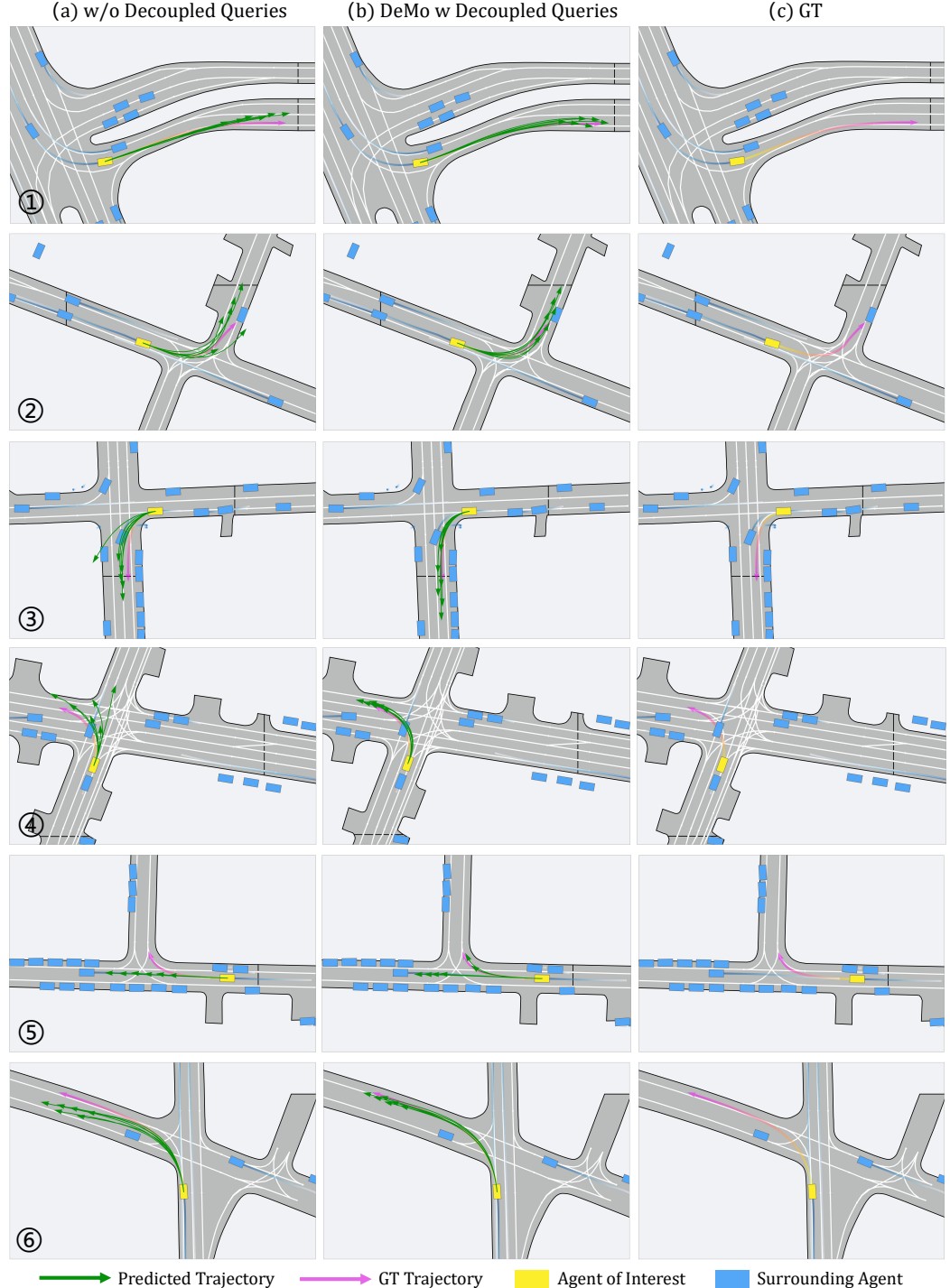

Figure 5: More qualitative results.

