# OpenReview forum: "DeMo: Decoupling Motion Forecasting into  Directional Intentions and Dynamic States"
_NeurIPS.cc/2024/Conference — NeurIPS 2024 poster_

### Official Review · Reviewer_Rxa7 · 2024-07-09

**Soundness:** 3
**Presentation:** 3
**Contribution:** 3
**Rating:** 5
**Confidence:** 4

**Summary:**

In this paper, a decoupled decoding process integrating intention and future state querying is proposed for motion prediction task. Through hybrid design with attention and Mamba, the proposed DeMo framework achieved strong performance on AV2 and NuScenes benchmarks.

**Strengths:**

1. A novel decoupled query design for intentions and future states.
2. A good trial in leveraging Mamba structure for efficient statewise decoding.
3. Solid performance on prediction benchmarks.

**Weaknesses:**

1. An over claim for SOTA performance: For instance, the LOF [1] method showcases much better performance in AV2 compared with proposed DeMo. I think a thorough comparison and corrections should be conducted.

2. Heavy computational load for long-horizon state queries.

**Questions:**

1. Whats is the computational cost such as FLOPs and training time of DeMo compared to other methods?

**Limitations:**

N/A.

---

> ### Author Rebuttal · Authors · 2024-08-06
>
> We thank the reviewer for the detailed review and the suggestions for improvement. Below are our responses to the reviewer’s comments:
>
> **Q1: Claim for SOTA performance.**
>
> Thank you for your attention. Regarding the Contemporaneous Work LOF [1], it was released on arXiv on June 20, 2024, after we submitted our paper to NeurIPS in May 2024. We will add a discussion on LOF to provide a thorough comparison in the revised paper.
>
> **Q2: Heavy computational load for long-horizon state queries.**
>
> To address this issue, we propose balancing efficiency and performance by utilizing one state query to represent several time steps, as shown in the left part of Table 6. This can effectively reduce the computational load.
>
> **Q3: Computational cost compared to other methods.**
>
> Thank you for this valuable suggestion. We have added a table below to further compare computational cost. The experiments are conducted on Argoverse 2 using 8 NVIDIA GeForce RTX 3090 GPUs.
>
> |Method|FLOPs|Training time|Memory|Parameter|Batch size|
> |:---:|:---:|:---:|:---:|:---:|:---:|
> |SIMPL [2]|19.7 GFLOPs|8h|14G|1.9M|16|
> |QCNet [3]|53.4 GFLOPs|45h|16G|7.7M|4|
> |DeMo (Ours)|22.8 GFLOPs|9h|12G|5.9M|16|
>
> > [1] FutureNet-LOF: Joint Trajectory Prediction and Lane Occupancy Field Prediction with Future Context Encoding. arXiv preprint: 2406.14422, 2024.
>
> > [2] SIMPL: A Simple and Efficient Multi-agent Motion Prediction Baseline for Autonomous Driving. IEEE Robotics and Automation Letters, 2024.
>
> > [3] Query-centric trajectory prediction. CVPR, 2023.

---

> > ### Comment · Reviewer_Rxa7 · 2024-08-12
> >
> > Thanks very much for perfectly addressing my problems. I would like to raise the Soundness, Contribution and maintaining the general rating.

---

> > > ### Author Response · Authors · 2024-08-13
> > >
> > > Dear Reviewer Rxa7
> > >
> > > We appreciate the reviewer's time for reviewing and thanks again for the valuable comments and the positive score!
> > >
> > > Best wishes
> > >
> > > Authors

---

### Official Review · Reviewer_V1LD · 2024-07-10

**Soundness:** 3
**Presentation:** 3
**Contribution:** 3
**Rating:** 6
**Confidence:** 4

**Summary:**

The manuscript presents a novel decoupling method for motion forecasting tasks, where the directional intentions are predicted first and the dynamic states following the predicted direction are predicted accordingly. The proposed solution is easy-to-follow, the model size is small, and the experimental results are convincing, achieving the first place of the Argoverse 2 dataset challenge assumed by the reviewer.

**Strengths:**

1. Simple model design with good performance
2. Extensive experiment and convincing results.

**Weaknesses:**

1. Some unclear details about loss function and model design.

**Questions:**

1. The loss function is simply a sum of all loss terms. Is it possible to have imbalanced losses, or in other words, one loss dominating the training?

2. Is there a Typo at Line 154? $Q_a$ or $Q_s$?

3. Between Lines 135-136, the authors said that $T_s$ and $T_f$ can differ. In this case, is the prediction conducted multiple times to fill the $T_f$ steps? If so, can the directional intention change in this process? If this is possible, can the model handle it?

**Limitations:**

See above.

---

> ### Author Rebuttal · Authors · 2024-08-06
>
> We thank the reviewer for the detailed review and the suggestions for improvement. Below are our responses to the reviewer’s comments:
>
> **Q1: About loss function.**
>
> Thank you for your careful consideration of our work. As indicated in the right part of Table 5 in our paper, each loss component in our model contributes effectively to the training process, ensuring that no single loss term dominates the optimization. This approach aligns with recent works, such as QCNet [1] and MTR [2], which also employ a simple sum of all loss terms.
>
> **Q2: A Typo at Line 154.**
>
> Yes, we apologize for the typo. It should be $Q_s$. We have revised it in the manuscript.
>
> **Q3: About $T_s$ and $T_f$ between Lines 135-136.**
>
> If $T_s$ and $T_f$ differ, it results in the cases mentioned in the left part of Table 6, where we reduce the number of state queries to make the model more efficient. For example, in Argoverse 2, with $T_f=60$ and $T_s=30$, this corresponds to the case in the third row of the left part of Table 6, where one state query represents two contiguous time steps. There is no need for multiple predictions, and the directional intentions remain unchanged.
>
> > [1] Query-centric trajectory prediction. CVPR, 2023.
>
> > [2] Motion transformer with global intention localization and local movement refinement. NeurIPS, 2022.

---

> > ### Comment · Reviewer_V1LD · 2024-08-12
> >
> > The authors have addressed most of my concerns. I will maintain my current score.

---

> ### Author Response · Authors · 2024-08-12
>
> Dear Reviewer V1LD
>
> We appreciate the reviewer's time for reviewing and thanks again for the valuable comments.
>
> Best wishes
>
> Authors

---

### Official Review · Reviewer_ZMh7 · 2024-07-11

**Soundness:** 3
**Presentation:** 3
**Contribution:** 3
**Rating:** 7
**Confidence:** 4

**Summary:**

DeMo: Decoupling Motion Forecasting into Directional Intentions and Dynamic States introduces a state of the art model architecture for motion forecasting (predicting the future trajectories of road actors for the purpose of autonomous driving).  The authors make two notable contributions.  First they provide an alternative to the typical one-query one-trajectory paradigm of current high performing models.  Instead they decompose queries into "mode queries" which attempt to capture directional intentions and "state queries" which attempt to capture the dynamic properties of a trajectory.  The authors also replace the sequence processing portion of the encoder (typically cross attention layers in current models or RNNs in older models) with Mamba blocks.  The authors include numerous ablations demonstrating that each of these contributions plays a substantive role in the performance of their model.  The authors present results on two significant motion forecasting benchmarks.

**Strengths:**

This is a very strong and well-presented manuscript.  Some strengths include:

### Presentation
- The copy is excellent.  The paper is well written and largely quite clear.
- Tables are excellent.
- Visual examples (figures 3-5) are clear and compelling.

### Results
The results are presented on two (of the three) most important motion forecasting benchmarks for self-driving.  In both cases DeMo is clearly SOTA.  They additionally present ensemble results.

### Analysis
The authors conduct an extensive set of ablation experiments.  They conduct the expected ablations (disabling various parts of their architecture).  Additionally, they explore the impact of: layer types (RNN vs Uni MB vs Bi MB), the number of layers, the number of state queries, the auxiliary losses.

**Weaknesses:**

This is already quite a solid manuscript, however I hope that the authors can selectively address a few of the weaknesses and questions below to arrive at an even better paper.

### Results
While AV2 and NuScenes are both important benchmarks, WOMD (Waymo) is probably the most important (or co-most-important) benchmark in this space right now.  It would have been nice to see results on WOMD.  There are tools like https://github.com/vita-epfl/UniTraj now to make this easier.

### Analysis
- While tables 4 and 5 clearly demonstrate that the query decomposition and auxilary losses contribute significantly to the model performance, there is little evidence to support the motivating intuitions.  For example, the introduction of state queries and the associated auxiliary loss is supposed to produce better dynamics.  Can we come up for a metric to measure that?  Can we find ways to demonstrate that our decomposition works as we expect?  Directional information is predominantly stored in mode queries while dynamics are encoded in state queries?
- I would have liked to see the authors get an RNN based model to successfully converge.  (The GRU result is disappointing).


### Reproducibility
The authors promise to release code (which would entirely alleviate my concern here), however I do not feel that the model is reproducible from the manuscript alone.  There is very little information about the auxilary losses.  Additionally there is no supplementary diagram or table with an explicit architectural description (layer sizes, normalization, etc).

**Questions:**

- Can you provide a precise formulation of the auxilary losses?
- Can you report training time?
- How does the model in ID3 (table 4) work?  Are the decouple queries just concatenated and fed directly to the MLP decoder?
- How does the model size vary across your ablations (table 4)?  I.e. why should I ascribe the performance improvements to architectural choices and not just growing model capacity?
- For inference speed can you report 99th percentile (or similar) rather than mean?  The AV industry is primarily concerned with worst-case performance.
- How does the multi-agent set up work?  Do I re-encode the scene in each agent-centric frame?
- Line 488 -- the text in this paragraph makes it sound like ZOH is a continuous->discrete transformation.  I might be wrong here, but I typically think of it as a digital->analog transformation.  Are we really doing something like the inverse of ZOH?

**Limitations:**

No concerns.

---

> ### Author Rebuttal · Authors · 2024-08-06
>
> We thank the reviewer for the detailed review and the suggestions for improvement. Below are our responses to the reviewer’s comments:
>
> ## _Response to Weaknesses._
>
> **1. Results: Results on WOMD (Waymo).**
>
> We provide results on WOMD using the settings in UniTraj [1], as shown below. The results of other methods are also from UniTraj.
>
> |Method|$minFDE_6$|$minADE_6$|
> |:---:|:---:|:---:|
> |MTR|1.78|0.78|
> |Wayformer|1.65|0.73|
> |AutoBot|1.65|0.73|
> |DeMo (Ours)|1.59|0.75|
>
> **2. Analysis: Better measure query decomposition.**
>
> Thank you for this valuable suggestion. We measure the outputs of state queries and mode queries with $minADE$ and $minFDE$, as shown below. We can see that the $minADE_1$ and $minFDE_1$ of the trajectories from state query outputs are better than those from mode query outputs. This means state dynamics are encoded in state queries. Additionally, there are six output trajectories from mode queries, indicating that directional information is predominantly stored in mode queries. The final outputs take advantage of the strengths of both. We will add this analysis to the revised paper.
>
> |Method|$minFDE_1$|$minADE_1$|$minFDE_6$|$minADE_6$|
> |:---:|:---:|:---:|:---:|:---:|
> |state query outputs|3.84|1.52|-|-|
> |mode query outputs|4.12|1.63|1.31|0.67|
> |final outputs|3.93|1.54|1.24|0.64|
>
> **3. Analysis: About RNN-based model and the GRU result.**
>
> As shown in the left part of Table 5, we conducted an ablation study on an RNN-based model to compare GRU with Mamba. If we directly replace Mamba with GRU to process state queries, it is difficult to achieve convergence, leading to rather poor results, as shown below. Perhaps RNN-based models can process state queries in some other way; this could be a research problem on its own.
>
> ||$minFDE_6$|$minADE_6$|$MR_6$|
> |:---:|:---:|:---:|:---:|
> |GRU|1.842|0.923|0.274|
>
> **4. Reproducibility: Architectural description.**
>
> Thank you for your question. The layer sizes are provided in Table 7 in the Appendix. For normalization, we use nn.LayerNorm, and for activation, we use nn.GELU. Additional details can be found in the implementation section at Lines 184 and 459.
>
>
>
> ## _Response to Questions._
>
> **1. A precise formulation of the auxiliary losses.**
>
> As in Line 143, we use an MLP to decode state queries into a single future trajectory $Y_f$ and calculate the loss with ground truth $Y_{gt}$ to obtain $L_{ts}$.
>
> $L_{ts} = {\rm SmoothL1}(Y_f, Y_{gt})$
>
> As in Line 150, we use MLPs to decode the future trajectories $Y_f$ and probabilities $P_f$. So $L_m$ is shown below:
>
> $Y_{best}, P_{best} = {\rm SelectBest}(Y_f, Y_{gt})$
>
> $L_{m} = {\rm SmoothL1}(Y_{best}, Y_{gt}) + {\rm CrossEntropy}(P_f, P_{best})$
>
> **2. Training time.**
>
> About 9 hours in total. Our settings are as mentioned in Line 184 and Line 463.
>
> **3. The model in ID3 (Table 4).**
>
> Yes, you are right. Decoupled queries are just concatenated and fed directly to the MLP decoder, and we use two auxiliary losses to optimize the two types of queries.
>
> **4. Model size vary across your ablations (Table 4).**
>
> The table below shows the model size variations across our ablations in Table 4. Additionally, we perform an ablation on the depth of Attention and Mamba layers, as shown in the right part of Table 6. We can see that even a single layer can achieve decent performance. This indicates that the performance improvements are due to architectural choices rather than merely increasing model capacity.
>
> |ID|model size|
> |:---:|:---:|
> |1|2.0M|
> |2|2.0M|
> |3|2.1M|
> |4|5.8M|
> |5|5.9M|
>
> **5. For inference speed.**
>
> Thank you for this valuable suggestion. In most scenarios ($>$ 90\%), the inference speed is between 25 to 45 ms. In some complex scenarios, it can go up to 70 ms, while in some easy scenarios, it is only 20 ms.
>
> **6. Multi-agent setting.**
>
> We use queries for each agent to predict their trajectories in the ego agent's coordinate system, so we do not re-encode the scene in each agent-centric frame. This approach avoids making the model costly.
>
> **7. For Line 488 about ZOH in Mamba.**
>
> ZOH can discretize continuous signals; the specific formula can be found in the Mamba [2] paper.
>
> > [1] UniTraj: A Unified Framework for Scalable Vehicle Trajectory Prediction. arXiv preprint:2403.15098, 2024.
>
> > [2] Mamba: Linear-Time Sequence Modeling with Selective State Spaces. arXiv preprint:2312.00752, 2023.

---

### Official Review · Reviewer_pCB4 · 2024-07-13

**Soundness:** 3
**Presentation:** 2
**Contribution:** 2
**Rating:** 5
**Confidence:** 4

**Summary:**

The paper presents DeMo, a novel framework for motion forecasting in autonomous driving systems. DeMo decouples the motion forecasting task into two distinct components: mode queries for capturing directional intentions and state queries for modeling dynamic states over time. This separation allows DeMo to separately optimize for multi-modality and dynamic state evolution, leading to a more comprehensive representation of future trajectories. The framework employs a combination of Attention and Mamba techniques for global information aggregation and state sequence modeling. Extensive experiments on the Argoverse 2 and nuScenes benchmarks demonstrate that DeMo achieves state-of-the-art performance in motion forecasting.

**Strengths:**

The overall idea behind DeMo is reasonable and technical soundness. Additionally, the experiments are comprehensive, demonstrating the results of DeMo across two different datasets, Argoverse 2 and nuScenes.

**Weaknesses:**

There are two major weaknesses: 1. Technical Descriptions Lack Clarity: The technical explanations of the methods and algorithms used in DeMo are not sufficiently clear, making it challenging for readers to fully understand the proposed techniques and their implementations. 2. Unconvincing Contributions: The claimed contributions of the paper are not entirely convincing. For instance, the paper doesn’t compare DeMo to models that have superior performance on existing leaderboards, questioning the novelty of its improvements. Additionally, there are previous works (e.g. Motion Mamba by Zhang et.al.) that have already combined Mamba and Transformer techniques for time series motion data, which undermines the claim that DeMo introduces a novel approach.

**Questions:**

1. In the first line of Eq. (2), what do the symbols {$\{ t_1, t_2, ...t_{T_s} \}$} represent, and how do $T_s$ and $T_f$ differ from or relate to each other? How are these time values obtained and utilized in the model?

2. For the motion model $Q_m$, what specific features are considered for different motion models? The paper does not clearly specify the features used in these motion models.

3. In Section 3.4, the loss function $L_{ts}$ is described as related to "intermediate features of time state." What exactly are these intermediate features, and what is their role in the model? This concept is not clearly defined in the paper.

4. Table 1 does not include baseline methods such as SEPT and SEPT++ that have better performance and earlier submission dates than DeMo. Similarly, Table 3 omits models like QCNet, which outperform DeMo. Why are these models not included in the comparisons?

5. The paper lacks detailed information about the model’s parameters and settings. Can you provide more specific details on the parameters used for DeMo?

**Limitations:**

The technical descriptions lack clarity and the overall contribution is limited.

---

> ### Author Rebuttal · Authors · 2024-08-06
>
> We thank the reviewer for the detailed review and the suggestions for improvement. Below are our responses to the reviewer’s comments:
>
> ## _Response to technical descriptions._
>
> **Q1: About Eq. (2) and related symbols meaning.**
>
> For the meaning of {$t_1,…,t_{T_s}$}: as in Line 135, these represent real-time differences of temporal states. For example, in Argoverse 2, we aim to predict future trajectories over 60 time steps. Therefore, if we use 60 state queries, $T_s=60$, and {$t_1,…,t_{T_s}$} corresponds to {$1,2,…,60$}.
>
> For $T_f$ and $T_s$: As stated in Line 114, $T_f$ refers to the future time steps. As stated in Line 135, $T_s$ represents the state steps used for initializing state queries. As stated in Line 233, in our default setting, $T_s=T_f$. We utilize one state query to represent several time steps in the left part of Table 6. In these cases, $T_s$ differs from $T_f$.
>
> **Q2: For the motion mode $Q_m$.**
>
> As stated in Line 146 and Line 151, we have $K$ mode queries to represent $K$ trajectories. We decode the $K$ future trajectories and their corresponding probabilities, using losses to optimize specific features for different motion modes, as done in QCNet [1], MTR [2], and other works.
>
> **Q3: About loss function $L_{ts}$ and intermediate features.**
>
> As stated in Line 143 and shown in Figure 2, we use an MLP to decode state queries into a single future trajectory $Y_f$ and calculate the loss with ground truth $Y_{gt}$ to obtain $L_{ts}$.
>
> Thus, the state queries $Q_s$ serve as intermediate features, similar to the mode queries $Q_m$. The role of these two types of intermediate features is to form the hybrid queries $Q_h$ to decode the final outputs.
>
> **Q4: About detailed information about the model’s parameters and settings.**
> - Training settings: Line 184, Line 459 (Appendix).
> - Dataset and metric settings: Line 175, Line 180.
> - Model size (5.9M) and inference speed (38ms): Line 242.
> - The number of layers in each component: Table 7 (Appendix).
>
> ## _Response to contributions._
>
> **Q5: Table 1 and Table 3 do not include baseline methods such as SEPT, SEPT++, QCNeXt.**
>
> For Table 1, our method (ranked 3rd) is better than SEPT [3] (ranked 4th) on the official leaderboard of Argoverse 2. SEPT is a self-supervised method utilizing all sets (including the test set) for pretraining. It is orthogonal to ours, and we believe that SEPT can also be integrated into our method for further improvements. Additionally, SEPT++ has not been released yet.
>
> For Table 3, our model is not specifically designed for the multi-agent setting, unlike QCNeXt [4], which is a method specifically designed for competition. We apologize for the insufficient comparison, which we have addressed in the revised paper.
>
> **Q6: Compare with previous works (e.g. Motion Mamba) for time series data.**
>
> Although both Motion Mamba [5] and our method utilize Mamba modules, the motivation and tasks are completely different. Our critical idea is decoupling motion forecasting, and Mamba is an effective tool to implement this aim. Additionally, the structures are also completely different. We will add a discussion to highlight the distinctions and contributions of our work compared to Motion Mamba in the revised manuscript.
>
> > [1] Query-centric trajectory prediction. CVPR, 2023.
>
> > [2] Motion transformer with global intention localization and local movement refinement. NeurIPS, 2022.
>
> > [3] Sept: Towards efficient scene representation learning for motion prediction. ICLR, 2024.
>
> > [4] Qcnext: A next-generation framework for joint multi-agent trajectory prediction. arXiv preprint:2306.10508, 2023.
>
> > [5] Motion mamba: Efficient and long sequence motion generation with hierarchical and bidirectional selective ssm. arXiv preprint:2403.07487, 2024.

---

> ### Author Response · Authors · 2024-08-12
>
> Dear Reviewer pCB4,
>
> We appreciate your time for reviewing, and we really want to have a further discussion with you to see if our response solves the concerns. We have addressed all the thoughtful questions raised by the reviewer (eg, technical descriptions and contributions), and we hope that our work’s impact and results are better highlighted with our responses. It would be great if the reviewer can kindly check our responses and provide feedback with further questions/concerns (if any). We would be more than happy to address them. Thank you!
>
> Best wishes,
>
> Authors

---

> > ### Comment · Reviewer_pCB4 · 2024-08-12
> >
> > Thank you for the reply, and I appreciate the additional information, which does help clarify some points for the reader. Regarding the methods on the leaderboard, I suggest including all relevant methods rather than selectively comparing with those that perform worse than the proposed model. Explaining why certain methods outperform the proposed model would not necessarily diminish the overall contribution of the paper. Additionally, it would be beneficial to address why one model performs better than another under different circumstances, as well as the limitations of the proposed method compared to others. Given the added details and improved clarity, I am inclined to raise my score.

---

> > > ### Author Response · Authors · 2024-08-13
> > >
> > > Dear Reviewer pCB4
> > >
> > > We appreciate the reviewer's time for reviewing and thanks again for the valuable comments and the improved score! We will revise and refine the paper as suggested in the revision.
> > >
> > > Best wishes
> > >
> > > Authors

---

### Decision · Program_Chairs · 2024-09-25

**Decision:**

Accept (poster)

**Comment:**

This paper introduces DeMo, a novel framework for motion forecasting in autonomous driving systems. DeMo decouples the motion forecasting task into two components: mode queries for capturing directional intentions and state queries for modeling dynamic states over time. The framework employs a combination of Attention and Mamba techniques for global information aggregation and state sequence modeling.

The reviewers generally agree that the paper presents an innovative approach with solid technical foundations. Reviewer ZMh7 notes that "This is a very strong and well-presented manuscript" and highlights the "excellent" presentation and comprehensive ablation studies. Reviewer V1LD praises the "Simple model design with good performance" and "Extensive experiment and convincing results."

The experimental results on Argoverse 2 and nuScenes benchmarks are seen as a strength by most reviewers. Reviewer ZMh7 states that "In both cases DeMo is clearly SOTA," while Reviewer V1LD mentions that the paper achieves "the first place of the Argoverse 2 dataset challenge."

However, there are some concerns and areas for improvement:
-  Clarity of technical descriptions: Reviewer pCB4 points out that "The technical explanations of the methods and algorithms used in DeMo are not sufficiently clear." This sentiment is echoed by Reviewer ZMh7, who notes that the model is not reproducible from the manuscript alone.
- Comparison with state-of-the-art models: Reviewer pCB4 and Reviewer Rxa7 both mention that the paper doesn't compare DeMo to some models with superior performance on existing leaderboards. Reviewer Rxa7 specifically mentions the LOF method as having "much better performance in AV2 compared with proposed DeMo."
- Computational cost: Reviewer Rxa7 raises concerns about the "Heavy computational load for long-horizon state queries" and asks for more information on the computational cost compared to other methods.
- Additional benchmarks: Reviewer ZMh7 suggests including results on the WOMD (Waymo) benchmark, noting it as "probably the most important (or co-most-important) benchmark in this space right now."
- Further analysis: Reviewer ZMh7 requests more evidence to support the motivating intuitions behind the query decomposition and auxiliary losses.

Recommendation: Despite these concerns, the overall sentiment from the reviewers is positive. Three out of four reviewers recommend acceptance (one "Accept" and two "Weak Accept"), while one reviewer gives a "Borderline Accept."

Based on these reviews, the recommendation is to accept the paper, contingent on the authors addressing the following key points in the camera-ready version:
- Improve the clarity of technical descriptions, especially regarding the auxiliary losses and architectural details.
Provide a more comprehensive comparison with state-of-the-art models, including those mentioned by the reviewers (e.g., LOF, SEPT, SEPT++, QCNet).
- Include information on computational costs and training time.
- If possible, include results on the WOMD benchmark or explain why it was not included.
- Provide additional analysis to support the intuitions behind the query decomposition and auxiliary losses.
- Address the specific questions raised by the reviewers, particularly those related to loss functions, model design, and handling of changing directional intentions.